# Galectins in Esophageal Cancer: Current Knowledge and Future Perspectives

**DOI:** 10.3390/cancers14235790

**Published:** 2022-11-24

**Authors:** Tesfay M. Godefa, Sarah Derks, Victor L. J. L. Thijssen

**Affiliations:** 1Department of Medical Oncology, Amsterdam UMC Location Vrije Universiteit Amsterdam, De Boelelaan 1117, 1081 HV Amsterdam, The Netherlands; 2Cancer Center Amsterdam, Cancer Biology & Immunology, De Boelelaan 1118, 1081 HV Amsterdam, The Netherlands; 3Oncode Institute, Jaarbeursplein 6, 3521 AL Utrecht, The Netherlands; 4Radiation Oncology, Amsterdam UMC Location Vrije Universiteit Amsterdam, De Boelelaan 1117, 1081 HV Amsterdam, The Netherlands; 5Laboratory for Experimental Oncology and Radiobiology, Center for Experimental and Molecular Medicine, Meibergdreef 9, 1105 AZ Amsterdam, The Netherlands

**Keywords:** squamous cell carcinoma, adenocarcinoma, gene expression, protein expression, diagnosis, prognosis, immunohistochemistry

## Abstract

**Simple Summary:**

The overall 5-year survival rate of esophageal cancer patients is poor. Galectins are glycan-binding proteins known to contribute to tumor initiation and progression. To get insight in the expression and potential function of galectins in esophageal cancer we performed a literature review. We found that galectins have been mainly studied in the context of esophageal squamous cell carcinoma and that galectin-1, -3, and -9 expression are most frequently reported. More research is required to provide better insights in the diagnostic, prognostic, and predictive value of galectins in esophageal cancer as well as their functional role in tumor progression

**Abstract:**

Esophageal cancer is a disease with poor overall survival. Despite advancements in therapeutic options, the treatment outcome of esophageal cancer patients remains dismal with an overall 5-year survival rate of approximately 20 percent. To improve treatment efficacy and patient survival, efforts are being made to identify the factors that underlie disease progression and that contribute to poor therapeutic responses. It has become clear that some of these factors reside in the tumor micro-environment. In particular, the tumor vasculature and the tumor immune micro-environment have been implicated in esophageal cancer progression and treatment response. Interestingly, galectins represent a family of glycan-binding proteins that has been linked to both tumor angiogenesis and tumor immunosuppression. Indeed, in several cancer types, galectins have been identified as diagnostic and/or prognostic markers. However, the role of galectins in esophageal cancer is still poorly understood. Here, we summarize the current literature with regard to the expression and potential functions of galectins in esophageal cancer. In addition, we highlight the gaps in the current knowledge and we propose directions for future research in order to reveal whether galectins contribute to esophageal cancer progression and provide opportunities to improve the treatment and survival of esophageal cancer patients.

## 1. Introduction

The esophagus is part of the digestive tract and facilitates the transport of solid and liquid substances from the oral cavity to the stomach. Similar to other organs, malignancies can develop in the esophagus and two histological types of esophageal cancer can be distinguished, i.e., esophageal squamous cell carcinoma (ESCC) and esophageal adenocarcinoma (EAC). As indicate by the name, ESCC originates from squamous epithelial cells and this subtype the most prevalent worldwide and is usually found in the upper/middle part of the esophagus. Adenocarcinoma is more frequently observed in the lower part of the esophagus and these cancers originate from (gastric) glandular cells [1,2]. The prospect for patients diagnosed with esophageal cancer is generally poor. The 5-year survival of patients with localized disease is approximately 45% which drops to 25% for patients with locoregional disease, and 5% for patients with metastatic disease. Standard treatment comprises neoadjuvant chemoradiotherapy (neoCRT) followed by surgical resection. Unfortunately, in only 1 out of 5 patients neoCRT results in a complete pathological response [3]. Since an incomplete response is a predictor of disease recurrence and poor patient survival [4], there is an urgent need to improve treatment efficacy.

Recently, adjuvant immunotherapy with nivolumab has been added to the standard treatment options for patients with an incomplete response to neoCRT [5]. The introduction of adjuvant immunotherapy exemplifies the importance of the tumor microenvironment in esophageal cancer. Indeed, we have already shown that the immune cell composition in the tumor microenvironment contributes to the treatment response in esophageal cancer patients [6,7]. There is also evidence for a role of tumor angiogenesis in esophageal cancer progression [8,9]. At the same time, effective development and implementation of therapies that target the tumor microenvironment require better insight in the mechanisms by which esophageal cancer cells shape the microenvironment to their benefit. 

Galectins represent a versatile glycan-binding protein family that is well-known for its tumor-promoting activity [10]. Aberrant expression of galectins is frequently observed in different cancer types and has been associated with tumor parameters like stage, grade and metastasis as well as with patient prognosis [11]. In the (tumor) microenvironment galectins have been linked to, e.g., angiogenesis stimulation and immune suppression (For extensive reviews, see [12,13,14,15]). For example, we described important roles of galectin-1 and galectin-9 in tumor angiogenesis [16,17,18]. With regard to immunomodulation, galectins have been described to induce immune evasion by, e.g., induction of T cell apoptosis [19,20] and stimulation of tolerogenic dendritic cell differentiation [21]. In addition, galectins are known to engage in glycan-mediated interactions with immune checkpoint proteins, like Tim-3, PD-1, and VISTA, thereby contributing to immune suppression [22,23]. In fact, galectins themselves have been suggested to serve as immune checkpoint proteins [24]. Based on all this, we anticipated that galectins are likely to contribute to disease progression in esophageal cancer. Surprisingly, the role of galectins in esophageal cancer is still poorly understood. In this review, we provide an overview of the current knowledge regarding the expression and function of galectins in esophageal cancer. We highlight the key outstanding questions and propose directions for future research in order to determine the function of galectins in esophageal cancer progression and to identify therapeutic opportunities to improve the treatment and survival of esophageal cancer patients.

## 2. The Galectin Protein Family

The galectin family consists of proteins that share a conserved carbohydrate-recognition domain (CRD) of approximately 130 amino acids which shows binding affinity for beta-galactosides [25]. The family name was coined in 1994 and at that time 4 members were listed, i.e., galectin-1 to -4. In addition, papers describing 3 putative members (galectin-5 to -7) were in preparation [25,26]. Currently, the family comprises 15 members which has allowed further classification into 3 different subtypes based on specific structural features, i.e., prototype (or dimeric), tandem-repeat, and chimera-type galectins (Figure 1a). The prototype galectins (galectin-1/-2/-5/-7/-10/-11/-13/-14/-15) consist of a single CRD which assembles into homo-dimers. Tandem-repeat galectins (galectin-4/-6/-8/-9/-12) have two distinct CRDs that are covalently linked by a short peptide sequence, while the only known chimera-type galectin (galectin-3) consists of a single CRD with an N-terminal non-lectin domain that contributes to protein oligomerization [15,27].

The capacity to form homodimers and multimers in combination with their glycan-binding activity allows galectins to serve as ‘bridging’ molecules that facilitate homotypic and heterotypic interactions between molecules and cells [15,28,29]. While such glycan-binding-related functions mostly occur in the extracellular environment, galectins can also engage in protein–protein interactions intracellularly, both in the cytoplasm as well as in the nucleus (Figure 1b). For example, in the cytoplasm galectin-1 dimers can directly interact with Raf-effectors which subsequently improves H-Ras nanoclustering in the membrane [30]. Recently, galectin-9 was suggested to interact with VAMP-3 in order to control cytokine trafficking in dendritic cells [31]. Haudek et al. provided an extensive overview of the intracellular binding partners of galectin-3, which includes nuclear proteins involved in splicing [32]. The same has been found for galectin-1 [33]. All these observations further add to the functional repertoire of galectins since such interactions have been shown to contribute to, e.g., intracellular signaling and splicing [28,33,34]. Finally, it is noteworthy to point out that galectins can form heterodimers with family members as well as with cytokines [35,36] and that they can be subjected to both post-transcriptional and post-translational modifications [37,38,39], aspects of which we are only beginning to understand the functional consequences. Nevertheless, it is nowadays well established that galectins constitute a protein family with versatile functions that contributes to a plethora of molecular, cellular and biological processes. Consequently, abnormal galectin expression and/or function has been linked to different pathologies, most notably cancer [11]. Their contribution to tumor initiation and progression was extensively summarized in a review by Girotti and coauthors describing the known roles of galectins in the different “hallmarks of cancer”, i.e., the seminal principles of tumorigenesis and tumor progression as defined by Hanahan and Weinberg [10,40]. Given their diverse functions in different cancer types, we performed a literature search to get insight in the role of galectins in esophageal cancer. The search identified 17 original research papers that reported on galectin expression and/or function in esophageal cancer cell lines, xenograft models and/or patient-derived material (Table 1). The majority of these studies involved the squamous subtype and most research was focused on galectin-1 and galectin-3. Below, the main findings of all these studies are discussed in more detail.

## 3. Galectins in Esophageal Cancer

### 3.1. Galectin-1

The first evidence of galectin expression in esophageal cancer was provided by Kayser and coworkers in 2001 [41]. The authors used an immunohistochemical approach to determine the expression of both galectin-1 and galectin-3 in 43 cancer tissues from ESCC patients. In addition, they used labeled galectins to explore the presence of galectin-binding glycans in the tumor tissue [41]. Galectin-1 expression was detected in 25/43 patients (±60%) while galectin-3 expression was detected in 33/43 patients (±80%). Galectin-1 and galectin-3 binding was observed in 35/43 and 27/43 cases, respectively. While galectin binding was not associated with tumor stage or nodal involvement, galectin-1 expression was less frequently observed in lymph node positive vs. lymph node negative patients (33% vs. 71%, *p* < 0.05). In contrast, Li et al. did not find a significant correlation of galectin-1 positivity with nodal stage after immunohistochemical analysis of expression in tumor tissues obtained from 93 ESCC patients [42]. Roughly half of the patients showed moderate to high expression and the other half showed no or low expression [42]. None of the evaluated clinicopathological parameters, including stage, grade or tumor location, was significantly associated with low (sum of absent/low score) or high (sum of moderate/high score) galectin-1 expression levels. At the same time, survival analysis showed that high galectin-1 expression was significantly correlated with reduced overall survival and disease-free survival. Multivariate analyses confirmed high galectin-1 expression as an independent prognostic factor for overall survival (HR = 2.1, *p* = 0.001) as well as for disease-free survival (HR = 1.8, *p* = 0.01) [42]. Thus, it appears that galectin-1 expression might have prognostic value in ESCC patients but this should be confirmed in larger patient groups.

Hoshino and coworkers explored whether the levels of circulating autoantibodies against galectin-1 could be a prognostic biomarker in ESCC [43]. For this, blood serum of 85 ESCC patients was analyzed. In addition, TCGA expression data from 184 ESCC patients were included. Galectin-1 autoantibodies were detected in approximately 10% of patients which was confirmed in a follow up study including more esophageal cancer cases as well as other tumor types [43,44]. While circulating autoantibodies and galectin-1 gene expression was significantly higher in patients vs. healthy controls, there was no significant difference observed with regard to prognosis [43]. Thus, while elevated expression in ESCC tumor tissues might underlie an increase in galectin-1 autoantibodies, the presence of these antibodies did not appear to be a valuable prognostic biomarker. Of note, when galectin-1 was included in a panel with three other autoantibodies, the diagnostic value was comparable to a combination of the classical tumor markers CEA (carcinoembryonic antigen) and SCC-Ag (squamous cell carcinoma antigen), i.e., 32% and 40% sensitivity, respectively [43]. Whether circulating levels of galectin-1 protein itself rather than autoantibodies have diagnostic or prognostic value in esophageal cancer still needs to be established. In that regard, in other tumor types, like colorectal cancer and glioma, it has been found that galectin-1 serum levels can have diagnostic and predictive value [11,58,59]. Moreover, we recently described that increased galectin-1 serum levels might serve as a negative predictive biomarker for the response to regorafenib and paclitaxel in patients with advanced esophagogastric cancer (EGC) that were refractory to first-line chemoradiotherapy [60]. Thus, it would be worthwhile to further explore the value of serum galectin-1 as a diagnostic, prognostic or even predictive biomarker in both ESCC and EAC.

Regarding the mechanism(s) responsible for elevated galectin-1 expression, a link has been made with Pituitary Tumor-Transforming Gene 1 (PTTG1) expression. Previously, it was found that PTTG1 is overexpressed in >60% of ESCC patients [45,61]. In addition, increased PTTG1 expression has been correlated to a higher metastatic potential of ESCC cell as it enhanced cell motility and increased the number of lymph node metastases in a mouse model [61]. Of note, high PTTG1 expression was not significantly associated with shorter overall survival [62]. Interestingly, Yan et al. found that increased PTTG1 expression actually induced the expression and secretion of galectin-1 via c-Myc. In addition, the elevated galectin-1 expression was involved in the increased motility and metastatic potential of ESCC cells [45]. 

More recently, it was described that a long non-coding RNA, i.e., ESCCAL-1 (ESCC-associated lncRNA), interacts with galectin-1 [46]. The interaction hampers ubiquitination of galectin-1 by SMURF1, an E3 ubiquitin-protein ligase. Consequently, elevated ESCCAL-1 expression in ESCC cell lines and patients was associated with higher galectin-1 protein levels. This resulted in enhanced cell proliferation through galectin-1-mediated cell cycle progression [46]. Thus, the increased levels of galectin-1 in ESCC could in part be related to induction of expression as well as to reduced protein turnover. Since galectin-1 expression is also known to be induced by cytokines [12,63] and under pro-angiogenic conditions [17] it is worthwhile to further explore which factors contribute to the observed high protein levels in ESCC patients.

Overall, current evidence suggests that galectin-1 is involved in ESCC progression which is in line with other tumor types. This warrants further investigation into the exact regulation and function of galectin-1, not only in ESCC but also in EAC. 

### 3.2. Galectin-3

As mentioned above, Kayser and colleagues were also the first to report on galectin-3 expression in ESCC. However, in contrast to galectin-1, the authors did not find any association between galectin-3 positivity and nodal involvement [41]. Another group also used immunohistochemical staining to evaluate whether galectin-3 was a prognostic marker for esophageal cancer patients. The study comprised 63 patients (62 ESCC and 1 EAC) with locally advanced esophageal cancer that received neoadjuvant chemoradiotherapy [47]. High expression was observed in 18 cases (29%) but no association with clinicopathological parameters or survival outcome was observed [47]. Similar observations were made in a study that included 154 ESCC tissues from patients that were treated by surgical resection without neoadjuvant treatment [48]. In this study, the authors also made a distinction between nuclear and cytoplasmic galectin-3 staining. In line with the previous findings, high cytoplasmic galectin-3 levels (*≥*45% positive cells, 72 cases) showed no significant association with clinicopathological parameters [48]. Furthermore, while high nuclear levels (*≥*30% positive cells, 23 cases) inversely correlated with vascular invasion and histological differentiation, neither cytoplasmic nor nuclear galectin-3 was prognostic [48]. Thus, the role of galectin-3 in ESCC appears to be minor.

Despite the limited diagnostic and prognostic value of galectin-3 in esophageal cancer, Balasubramanian and colleagues included 52 esophageal carcinoma patients in a study that evaluated whether galectin-3 levels in urine could be used to monitor disease status and/or treatment efficacy [49]. Compared to healthy controls, urine galectin-3 levels were significantly higher in esophageal cancer patients, most notably in patients with metastatic disease. In fact, galectin-3 levels significantly correlated with disease stage which is different from the observations reported in esophageal tumor tissues [49]. However, the study using urine samples did not specify the types of esophageal cancer so these opposing findings might reflect a difference between ESCC and EAC which needs further investigation.

Regarding the functional role of galectin-3 expression in esophageal cancer, Liang and colleagues studied the effect of increased galectin-3 expression on ESCC cell behavior. For this, the authors virally transduced an ESCC cell line (Eca-109) in order to overexpress galectin-3 [50]. Subsequently, different functional features of parental and galectin-3 overexpressing cells were compared, including proliferation, apoptosis, migration and invasion. The parental cells already showed considerable galectin-3 protein expression and in the transduced cells this level increased only ±1.5 fold. Nevertheless, the transduced cells displayed significantly lower apoptosis and an increased proliferative, migratory and invasive capacity [50]. Thus, despite the limited diagnostic or prognostic value, galectin-3 might be involved in regulation of ESCC progression and present as a target for treatment. The latter was explored by the same group in a follow-up study. Using the same cell line and an siRNA approach, the authors were able to significantly inhibit galectin-3 protein levels by 65 to 90% [51]. An inhibitory effect of knockdown on proliferation became apparent 72 h after siRNA transfection. In addition, migration and invasion were hampered while apoptosis levels increased following galectin-3 knockdown [51]. Both studies suggest that galectin-3 expression could serve as a potential therapeutic target in ESCC. In line with this, Cui et al. investigated the effect of targeting galectin-3 in the context of resistance to EGFR-targeted therapy with gefitinib [52]. The authors found that siRNA-mediated knockdown of galectin-3 sensitized the ESCC cell line TE-8 to gefitinib treatment. This was associated with reduced internalization of EGFR in the knockdown cells. This was confirmed in an in vivo tumor model, in which galectin-3 knockdown sensitized tumors to gefitinib to a similar extend as inhibition of dynamin-dependent endocytosis [52]. Finally, Xu and colleagues found that synephrine, a compound isolated from citrus tree leaves, effectively inhibited the proliferation, migration, invasion and colony formation of 2 ESCC cell lines (KYSE30 + KYSE270) [53]. The compound also hampered in vivo tumor growth in a xenograft mouse model and sensitized cells to 5-FU treatment. Importantly, subsequent proteomic analyses indicated reduced AKT and ERK signaling as key effects with galectin-3 as the upstream regulatory protein [53]. In line with this, synephrine treatment reduced galectin-3 expression in the ESCC cells [53]. Together with the findings by Cui et al., this suggests that targeting galectin-3 could be beneficial for ESCC patients as it has direct effects on the tumor cells and sensitizes the cells for combination treatment. 

Collectively, despite the limited diagnostic and prognostic value of galectin-3 in ESCC, the protein might serve as a therapeutic target, in particular in combination with other therapeutics in this subtype. However, in line with galectin-1, hardly anything is known regarding galectin-3 in EAC and more research is required to establish the therapeutic value and function in both subtypes.

### 3.3. Galectin-7

A possible role of galectin-7 in esophageal cancer has been reported by only 1 study. Zhu and coworkers observed higher galectin-7 protein levels in ESCC tissue after proteomic analyses of patient-matched normal and tumor samples [54]. The elevated expression was confirmed in independent sets of patient samples which showed high expression in more than 50% of all cases. Overall, an approximately 2.5-fold increase was found in tumor vs. normal tissue. In addition, it was observed that galectin-7 protein was detectable in the nucleus, cytoplasm and on the membrane of cells [54]. Interestingly, within the ESCC samples, well differentiated (grade I) tumors expressed significantly higher levels of galectin-7 as compared to poorly differentiated (grade III) tumors. Of note, using a comparable proteomic approach, Qi et al. did not identify increased galectin-7 expression in ESCC [64]. This might be related to experimental setup and/or specific protein spot selection after 2D gel electrophoresis but it also indicates that deciphering the role of galectin-7 in ESCC requires further studies.

### 3.4. Galectin-9

Regarding the role of galectin-9 in esophageal cancer, Hou and coworkers analyzed the expression of galectin-9 and its binding partner Tim-3 in tumor tissues obtained from 45 ESCC patients during curative surgical resection. None of the patients received neo-adjuvant treatment [55]. Based on immunohistochemical staining, 7 patients were classified as ‘galectin-9 high’ while 38 were classified as ‘galectin-9 low’. Of the latter, 11 cases did not show any galectin-9 staining at all. While the galectin-9 expression was not significantly associated with any clinicopathological parameters, survival analyses showed that patients in the galectin-9 high group had significantly longer disease-free survival as compared to patients in the galectin-9 low group. However, subsequent multivariate Cox regression analysis did not identify galectin-9 as an independent prognostic factor in this patient cohort [55]. Since the number of patients included in this study was limited, additional research is required to determine whether galectin-9 has any prognostic value in ESCC. Of note, there was a negative correlation between galectin-9 expression and Tim-3 expression [55].

The effect of exogenously added galectin-9 on different esophageal cancer cell lines has also been explored. In EAC cell lines, galectin-9 treatment dose-dependently inhibited tumor cell proliferation and at the same time induced caspase-independent apoptosis [56]. Further research focused on a single cell line and indicated that galectin-9 might inhibit the autophagy flux, while the effects on cell cycle regulatory proteins and cell cycle progression were minimal and absent, respectively [56]. Apparently, galectin-9 inhibits EAC cell growth mainly via induction of apoptosis. Of note, the authors observed that galectin-9 also induced CXCL8 (IL-8) secretion by the cancer cells. In ESCC, increased CXCL8 expression was previously shown to be a predictive marker for poor prognosis [65], so it is tempting to speculate that this is also related to galectin-9 expression. At the same time, since galectin-9 hampers tumor growth it is difficult to reconcile galectin-9 induction of CXCL8 with poor prognosis. Nevertheless, in a follow-up study by the same group, the growth inhibitory and pro-apoptotic effects of galectin-9 were confirmed in esophageal squamous cell carcinomas using different in vitro and in vivo models [57]. Moreover, it was suggested that apoptosis was induced via activation of the intrinsic pathway, i.e., mitochondria-mediated apoptosis [57]. Unfortunately, CXCL8 expression or secretion was not assessed.

The data on galectin-9 are limited but suggest that this family member might serve as a prognostic marker and therapeutic protein in esophageal cancer. In EAC, the protein hampers tumor progression which is in line with the observation in ESCC that high galectin-9 expression is associated with better overall survival. These findings warrant further studies of this galectins in esophageal cancer.

## 4. Summary and Future Perspectives

Here, we evaluated the current knowledge with regard to galectin expression and function in esophageal cancer. This was instigated by the increasing insight that the tumor microenvironment plays an important role in the poor treatment response of esophageal cancer patients together with the known tumor-promoting roles of galectins in the tumor microenvironment. Based on the available literature, it can be concluded that galectins have been mainly studied in the context of ESCC. Within this subtype, galectin-1, -3, and -9 are most frequently reported which is in line with other cancer types [11]. At the same time, it is important to note that the published work comprises a broad spectrum of techniques (IHC, proteomics, RNA expression), sample types (cell lines, tissues, blood, urine, TCGA database), and readouts (protein levels, RNA levels, protein binding, auto-antibodies). Consequently, the findings are occasionally biased and difficult to compare since each approach has its own limitations regarding, e.g., sensitivity, specificity, and applicability. This makes it difficult to draw definite conclusions regarding the expression and function of galectins in ESCC. Thus far, no broad galectin expression analyses have been performed in ESCC, so it cannot be excluded that other galectins play a role in this subtype. For example, in tumors of the digestive tract, alterations have been reported in the expression of, e.g., galectin-4 [66,67,68] and galectin-8 [69,70]. In line with this, we recently found that focal amplification of the galectin-4 encoding gene LGALS4 (19q13.12-q13.2), was associated with poor OS and PFS in patients with advanced esophagogastric carcinoma that received regorafenib and paclitaxel treatment following first-line therapy [60]. Thus, future research should include a broader expression analysis of galectins in ESCC. This also applies to EAC, as the expression and role of galectins in this subtype is poorly explored. Using information from existing public data sets together with extensive profiling using novel techniques like single cell RNA sequencing should help to identify the key galectins related to both ESCC and EAC. Proteomic approaches can also shed light on the expression and secretion of galectins in esophageal cancer. Obtaining more comprehensive galectin expression signatures for both subtypes will provide more insight in the diagnostic, prognostic, or even predictive value of galectins in esophageal cancer. Of note, it is important to also determine the cellular localization of each galectin as this can affect the protein activity and function [11].

Insight in the expression and cellular localization of galectins in esophageal cancer cells will also provide clues to study how galectin contribute to the malignant behavior of esophageal cancer cells. As already mentioned earlier, several galectins have been linked to the key steps of tumorigenesis and tumor progression as described by Hanahan and Weinberg [10,40]. Indeed, we, and many others, have revealed essential functions of galectins in multiple hallmarks of cancer, including, sustaining proliferative signaling [30,71,72], avoiding of immune destruction [14,20,73], stimulating angiogenesis [16,74,75,76], activating invasion and metastasis [77,78,79], and resisting cell death [80,81,82]. In line with this, the current literature review indicates that galectins are also involved in multiple cellular functions of esophageal cancer cells, including cell proliferation/cell cycle progression [46,56], migration/invasion [45,53] and apoptosis [56,57]. At the same time, our understanding of the exact role of galectins in malignant esophageal cells is limited and confined to only a few galectins and a few cancer hallmarks. Given the diverse functionality of galectins it can be anticipated that these proteins contribute to esophageal cancer progression on multiple levels which needs further investigation.

Another aspect that requires more in-depth research relates to the main function of galectins, i.e., glycan binding. As galectins are able to decipher the ‘glycan codes’ that are presented by cells and the extracellular environment, alterations in esophageal cancer cell glycosylation could give insight in the effects that galectins exert on malignant cells or on other cells in the tumor microenvironment. For example, using lectin microarrays, Xia and colleagues found altered glycosylation patterns in ESCC compared to normal cells [83]. High binding of a specific lectin (LCA) was associated with different clinical parameters as well as with poor overall survival of esophageal cancer patients. In addition, enhanced LCA binding was associated with increased expression levels of the cell adhesion molecule CD146, a cell surface molecule which is known to bind galectin-1 and galectin-3 [78,84]. Previously, the same group found that increased expression of a specific glycosyltransferase (C1GalT1) was also associated with poor overall survival and radioresistance in ESCC [85]. This was linked to altered glycosylation of β1-integrin, another known ligand of different galectins [86,87,88,89]. Based on these examples, it can be anticipated that—similar to other cancers [10,90]—altered tumor cell glycosylation in combination with altered galectin expression contributes to tumor progression in esophageal cancer [91]. This warrants further studies into the (aberrant) mechanisms of glycosylation in esophageal cancer by, e.g., more extensive glycan profiling and analysis of the expression/function of enzymes involved in the glycosylation machinery. 

Better insight in the role of the galectin-glycosylation axis is particularly relevant for understanding how esophageal cancer cells can shape the tumor immune microenvironment to their benefit. It is well accepted that galectin-glycan interactions are key events in immunoregulation [14]. Consequently, galectins contribute to physiological immune homeostasis but also to pathological immune responses that occur during, e.g., malignant disease. Apart from interacting with immune checkpoint proteins, like Tim-3, PD-1, and VISTA [22,23] and acting as immune checkpoint proteins themselves [24], we and others have shown that galectins can also heterodimerize with cytokines which further adds to their immunomodulatory activity [92,93,94]. As already mentioned, the composition of the immune cell population within the tumor appears to affect how esophageal cancer patients respond to treatment [6,7]. In addition, there is ample evidence that the extracellular milieu contributes to esophageal cancer progression [95]. Moreover, as reviewed by Elola et al., galectins are known to be expressed by non-malignant cells that reside in the tumor microenvironment, including immune cells, endothelial cells, and fibroblasts [96]. This expression can contribute to tumor progression by stimulating tumor cell growth, tumor angiogenesis, immunosuppression, and tumor metastasis. Thus, it can be anticipated that galectin expression and secretion in the tumor microenvironment of esophageal cancer also contributes to disease progression. However, the exact role of galectins in the microenvironment of esophageal cancers is currently unknown and requires further investigation.

## 5. Conclusions

In conclusion, esophageal cancer is malignant disease with poor patient outcome. Novel treatment options and therapeutic targets are key to improving the prospects for esophageal cancer patients. Based on the current literature review we propose that the therapeutic value of galectins in esophageal cancer should also be further explored. Nowadays, several galectin-targeting compounds are available, some of which are already evaluated in clinical trials in a variety of cancer types, including, colon, lung, breast, head&neck, prostate, and melanoma [97,98]. Most trials involve phase 1/2 studies to test the safety of glycan-based or small molecule inhibitors that target galectin-1 and/or galectin-3 (For a recent overview, see Martin-Saldaña et al. [98]). In addition, an ongoing phase1/2 trial in patients with metastasized solid tumors explores the safety, pharmacokinetics, and anti-tumor activity of a galectin-9-targeting antibody, either as monotherapy or combined with chemotherapy or anti-PD-1 treatment (NCT04666688). Thus, the upcoming years will give insight in the applicability and efficacy of these treatments. Obviously, to determine whether galectin-targeted therapies might also be successful in esophageal cancer, it is vital to gain better insights in the expression and exact role of galectins in this malignant disease. With this review, the first step in that direction was taken by highlighting the gaps in the current knowledge and identifying the outstanding challenges. This will help to guide future research efforts that aim to improve the treatment and survival of esophageal cancer patients.

## Figures and Tables

**Figure 1 cancers-14-05790-f001:**
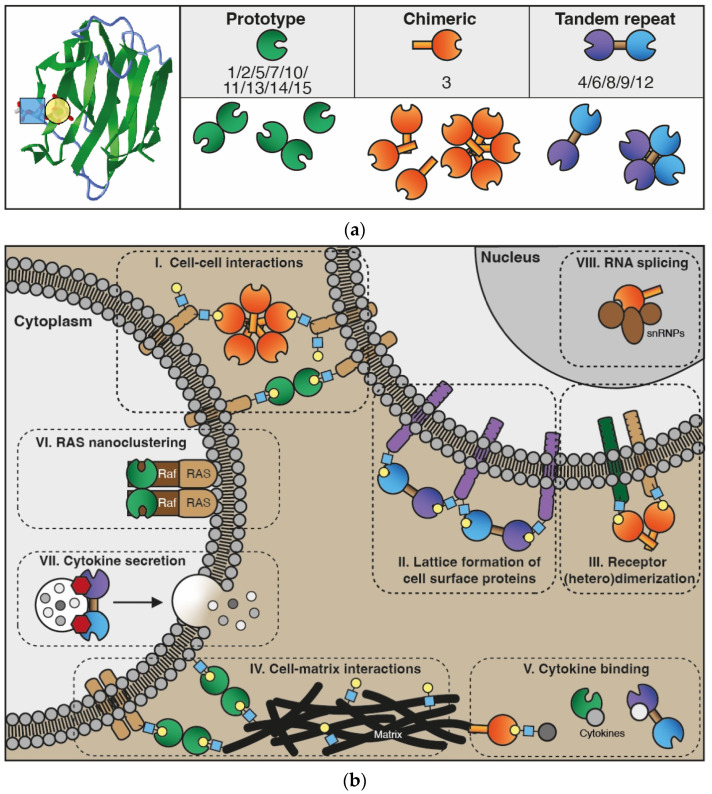
The galectin protein family. (**a**) Left panel: Cartoon of the anti-parallel beta-sheet structure forming the carbohydrate recognition domain of galectin-1 (in green). On the left, the interaction of a LacNAc (N-acetyllactosamine; blue-yellow) moiety in the binding groove is shown. Right panel: Overview of the 15 galectins that are expressed in humans. See text for explanation of the subgroups. (**b**) Cartoon depicting some of the key functions of galectins intra- and extracellularly. For more detailed information, refer to the reviews referred to in the main text. In the extracellular environment and on the cell surface, galectins can interact with glycoconjugates (blue-yellow) to facilitate, e.g., cell–cell interactions (I) and cell–extracellular matrix interactions (IV) to enable cell adhesion and migration. In addition, extracellular galectins can mediate interactions between molecules in the cell membrane to stimulate, e.g., lattice formation (II) and receptor dimerization (III), thereby promoting receptor cell surface retention and cell signaling. More recently, galectins were also found to heterodimerize with cytokines (V), affecting both galectin and cytokine activity. In the cytoplasm, galectins have been shown to engage in protein–protein interactions that facilitate, e.g., H-Ras nanocluster formation (VI) and signaling as well as cytokine secretion (VII). Finally, in the nucleus, galectins can interact with small nuclear ribonucleoproteins (VIII) thereby contributing to mRNA splicing.

**Table 1 cancers-14-05790-t001:** Galectins in esophageal cancer.

Subtype ^1^	Sample Type ^2^	Main Finding	Ref.
** *Galectin-1* **			
ESCC	Patients (*n* = 43)	Frequency of galectin-1 positive cells is lower in lymph node positive vs. lymph node negative patients.	[41]
ESCC	Patients (*n* = 93)	High galectin-1 expression is an independent prognostic factor for OS and DFS in ESSC.	[42]
ESCC	Patients (*n* = 85)	No prognostic value of circulating autoantibodies against galectin-1.	[43]
ESCC	Patients (*n* = 172)	No prognostic value of circulating autoantibodies against galectin-1	[44]
ESCC	Cell lines (EC9706/KYSE150)	Galectin-1 is induced by PTTG and stimulates cell motility.	[45]
ESCC	Patients (*n* = 41) Cell lines (EC9706/EC109/KYSE70/KYSE150/KYSE450), Xenografts	Interaction with long non-coding RNA ESCCAL-1 stabilizes galectin-1 protein and promotes cell cycle progression.	[46]
** *Galectin-3* **			
ESCC	Patients (*n* = 43)	No association of galectin-3 with nodal involvement or tumor stage.	[41]
ESCC	Patients (*n* = 63)	High galectin-3 expression is not prognostic for survival outcome.	[47]
ESCC	Patients (*n* = 154)	Nuclear or cytoplasmic galectin-3 expression is not prognostic in ESCC.	[48]
NS	Patients (*n* = 52)	Urinary galectin-3 is a potential diagnostic tool to monitor or follow up the disease stage.	[49]
ESCC	Cell line (Eca-109)	galectin-3 overexpression increases malignant behavior.	[50]
ESCC	Cell line (Eca-109)	Galectin-3 knockdown represents a therapeutic strategy for ESCC	[51]
ESCC	Cell lines (KYSE-450/TE-8)	Galectin-3 knockdown increases gefitinib sensitivity.	[52]
ESCC	Cell lines, Xenografts (KYSE30/KYSE270)	Galectin-3 inhibition of hampers esophageal tumor growth and metastasis.	[53]
** *Galectin-7* **			
ESCC	Patients (*n* = 50)	Galectin-7 as a potential biomarker for ESCC.	[54]
** *Galectin-9* **			
ESCC	Patients (*n* = 45)	Low galectin-9 expression is associated with a poor prognosis.	[55]
EAC	Cell lines (OE19/OE33/SK-GT4/OACM5.1c)	Galectin-9 suppresses proliferation and induces apoptosis in EAC cells.	[56]
ESCC	Cell lines, Xenografts (KYSE-150/KYSE-180)	Galectin-9 induces mitochondria-mediated apoptosis of esophageal cancer.	[57]

^1^ NS = Not Specified; ESCC = esophageal squamous cell carcinoma; EAC = esophageal. ^2^ In brackets the patient number or the cell lines used are indicated.

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
