# Peer review of "Galectins in Esophageal Cancer: Current Knowledge and Future Perspectives"

_cancers, 2022, doi:10.3390/cancers14235790_

Round 1
Reviewer 1 Report
Godefa et al., is presenting a short review article about current knowledge and future perspective of galectins in esophageal cancer. The authors briefly introduced galectins and summarized prior studies in the field. The manuscript is well written, and the structure is well constructed. However, there are few concerns and suggestions the reviewer here would like to address.
Major concerns:
1. In Figure 1, the authors illustrate the families and functions of galectins based on reference number 29, however, this diagram is vague and not sufficient to show the fundamental role of galectins. For example, Figure 1b does not clearly show the difference in between homodimer and heterodimer galectins in cellular functions. Also, the authors citate several references but all these information is not clearly present in Figure 1b. In addition, Figure 1b does not clearly show the differences in between the three types of galectins. There are review articles which show figures illustrating informative information of galectins as following list. The authors should remake this figure and provide informative detail about galectin families and their fundamental functions/roles in normal and disease examples.
https://www.nature.com/articles/nri2536
https://www.frontiersin.org/articles/10.3389/fimmu.2021.784473/full
https://www.frontiersin.org/articles/10.3389/fimmu.2012.00199/full
2. It is understandable that current knowledge of galectin in ESCC is not well studied and it’s limited. But there is a lack of opinion made by the authors. It would be beneficial for readers if the authors could add some perspectives based on their own opinion about would galectin be a critical molecule in ESCC diagnosis/treatment, and could we use public data sets to study galectins in ESCC.
3. There are publications showing galectins play roles in immune responses (For example, https://www.nature.com/articles/nri2536). The reviewer here would suggest the authors adding a separate section or a paragraph to discuss potential role of galectin in immune surveillance during ESCC progression and tumor metastasis.
4. The research papers that the authors have discussed here measured/detected galectins by IHC, immunoblot, RNA assay, autoantibody, TCGA public database, urine sample, glycan binding assay and glycan lectin array etc. However, the results/conclusions from these papers are bias and not comparable. It would be valuable if the authors could add a discussion to comment the approaches and limitations in detecting galectins or its target glycans.
5. I would suggest the authors adding name of cell lines were being used in each of prior studies of Table 1.
Author Response
We would like to thank the reviewer for the time and comments to improve the quality of the manuscript. Below a point-by-point response can be found
- In Figure 1, the authors illustrate the families and functions of galectins based on reference number 29, however, this diagram is vague and not sufficient to show the fundamental role of galectins. For example, Figure 1b does not clearly show the difference in between homodimer and heterodimer galectins in cellular functions. Also, the authors citate several references but all these information is not clearly present in Figure 1b. In addition, Figure 1b does not clearly show the differences in between the three types of galectins. There are review articles which show figures illustrating informative information of galectins as following list. The authors should remake this figure and provide informative detail about galectin families and their fundamental functions/roles in normal and disease examples.
https://www.nature.com/articles/nri2536
https://www.frontiersin.org/articles/10.3389/fimmu.2021.784473/full
https://www.frontiersin.org/articles/10.3389/fimmu.2012.00199/full
>> For clarity, we have updated/modified the figure in line with the suggested references and added the following text to give some additional background on the potential role of intracellular galectins:
While such glycan-binding-related functions mostly occur in the extracellular environment, galectins can also engage in protein-protein interactions intracellularly, both in the cytoplasm as well as in the nuclues (Figure 1b). For example, in the cytoplasm galectin-1 dimers can directly interact with Raf-effectors which subsequently improves H-Ras nanoclustering in the membrane (Blaževitš et al., 2016). Recently, galectin-9 was suggested to interact with VAMP-3 in order to control cytokine trafficking in dendritic cells (Santalla Méndez et al., 2022). Haudek et al. provided an extensive overview of the intracellular binding partners of galectin-3, which includes nuclear proteins involved in splicing (Haudek et al., 2010).
Of note, it is impossible and outside the scope of this review to comprehensively describe all the fundamental functions/roles of galectins in normal and diseased tissue. The current review provides already several relevant references to reviews which address this for the readership that wants more in-depth information.
- It is understandable that current knowledge of galectin in ESCC is not well studied and it’s limited. But there is a lack of opinion made by the authors. It would be beneficial for readers if the authors could add some perspectives based on their own opinion about would galectin be a critical molecule in ESCC diagnosis/treatment, and could we use public data sets to study galectins in ESCC.
>> To address this, we have added some conclusive lines at the end of several paragraphs and additional text to the 'summary and future perspectives' section.
- There are publications showing galectins play roles in immune responses (For example, https://www.nature.com/articles/nri2536). The reviewer here would suggest the authors adding a separate section or a paragraph to discuss potential role of galectin in immune surveillance during ESCC progression and tumor metastasis.
>> Indeed, galectins are well known for their immunomodulatory functions. This is already recognized in the current review by referring to key reviews that discuss these functions in the introduction. To our opinion, a detailed description of all these functions (an entire review on itself) is outside the scope of the current review, also because no information is available on immunosuppression by galectins in esophageal cancer. This is also further discussed in the 'summary and future perspectives' section of the current review. Nevertheless, to provide some examples, we have added the following text to the introduction:
For example, we described important roles of galectin-1 and galectin-9 in tumor angio-genesis (Aanhane et al., 2018; Thijssen et al., 2006; Thijssen et al., 2010). With regard to immunomodulation, galectins have been described to induce immune evasion by, e.g., induction of T cell apoptosis (Stillman et al., 2006; Toscano et al., 2007) and stimulation of tolerogenic dendritic cell differentiation (Ilarregui et al., 2009).
- The research papers that the authors have discussed here measured/detected galectins by IHC, immunoblot, RNA assay, autoantibody, TCGA public database, urine sample, glycan binding assay and glycan lectin array etc. However, the results/conclusions from these papers are bias and not comparable. It would be valuable if the authors could add a discussion to comment the approaches and limitations in detecting galectins or its target glycans.
>> Indeed, apart from the overall limited data that is available, the type of data and data collection is highly variable and sometimes biased. To make this aware to the readership, we have added the following text to the discussion section:
At the same time, it is important to note that the published work comprises a broad spectrum of techniques (IHC, proteomics, RNA expression), sample types (cell lines, tissues, blood, urine, TCGA database), and readouts (protein levels, RNA levels, protein binding, auto-antibodies). Consequently, the findings are occasionally biased and difficult to compare since each approach has its own limitations regarding, e.g., sensitivity, specificity, and applicability. This makes it difficult to draw definite conclusions regarding the expression and function of galectins in ESCC.
- I would suggest the authors adding name of cell lines were being used in each of prior studies of Table 1.
>> The requested information has been added to the table. In addition, we have also added the number of patients for studies that used patient samples.
Reviewer 2 Report
Godefa et. al. has provided a very comprehensive summary of the role of galectins in esophageal cancer.
If possible, expanding on glycan modifications in esophageal cancer and how these would possibly alter galectin binding will add value. The authors touch on galectins being used in clinical trials but expanding on this will add to the importance of this review as a go to source.
Author Response
We would like to thank the reviewer for the time and comments to improve the quality of the manuscript. Below a point-by-point response can be found.
If possible, expanding on glycan modifications in esophageal cancer and how these would possibly alter galectin binding will add value. The authors touch on galectins being used in clinical trials but expanding on this will add to the importance of this review as a go to source.
>> To our knowledge, all the available findings on altered glycosylation in esophageal cancer is already addressed in the section 'summary and future perspectives'. We have added the following sentence to that paragraph to highlight that more research is needed on this subject:
This warrants further studies into the (aberrant) mechanisms of glycosylation in esophageal cancer by, e.g., more extensive glycan profiling and analysis of the expres-sion/function of enzymes involved in the glycosylation machinery.
Regarding the clinical trials, we have added the following text to the 'summary and future perspectives section:
Nowadays, several galectin-targeting compounds are available, some of which are al-ready evaluated in clinical trials in a variety of cancer types, including, colon, lung, breast, head&neck, prostate, and melanoma (Wdowiak et al., 2018; Martin-Saldaña et al., 2022). Most trials involve phase 1/2 studies to test the safety of glycan-based or small molecule inhibitors that target galectin-1 and/or galectin-3 (For a recent overview, see Martin-Saldaña et al. (Martin-Saldaña et al., 2022)). In addition, an ongoing phase1/2 trial in patients with metastasized solid tumors explores the safety, pharmacokinetics, and anti-tumor activity of a galectin-9-targeting antibody, either as monotherapy or combined with chemotherapy or anti-PD-1 treatment (NCT04666688). Thus, the upcoming years will give insight in the applicability and efficacy of these treatments. Obviously, to determine whether galectin-targeted therapies might also be successful in esophageal cancer, it is vital to gain better insights in the expression and exact role of galectins in this malignant disease. With this review, the first step in that direction was taken by highlighting the gaps in the current knowledge and identifying the outstanding challenges.
Reviewer 3 Report
The paper is well written and balanced. Topic is interesting.
Author Response
The paper is well written and balanced. Topic is interesting.
>> We thank the reviewer for this positive comment.
Reviewer 4 Report
This review by Tesfay M Godefa etal. summarize the current literature with regard to the expression and potential functions of galectins in esophageal cancer. Although the summary about galectin studies is relatively comprehensive, there are a number of issues with this manuscript as listed below:
(1) As galectins are not only expressed in tumor cells but also in other immune cells, authors should conduct a comprehensive analysis of their expression in the whole tumor microenvironment. For example, modest expression of galectin 9 was found in both leukocytes (13%) and tumor cells (7%) in human PDA (doi:10.1038/nm.4314).
(2) The author simply listed several galectins including galectin-1,galectin-3,galectin-7,galectin-9 without in-depth analysis and outlook. Specially, most studies about functions of galectins in easophageal were inconsistent and contradictory. So will galectins be a good target for cancer therapy?
(3) In last paragraph, authors showed “several galectin-targeting compounds are available, some of which are already evaluated in clinical trials”. But the information about clinical trials is a little, can the authors describe it in more detail as it is very important for further study of galectins.
(4) There are still a few minor issues worth fixing: “in vitro and in vivo” should be italic.
Author Response
We would like to thank the reviewer for the time and comments to improve the quality of the manuscript. Below a point-by-point response can be found.(1) As galectins are not only expressed in tumor cells but also in other immune cells, authors should conduct a comprehensive analysis of their expression in the whole tumor microenvironment. For example, modest expression of galectin 9 was found in both leukocytes (13%) and tumor cells (7%) in human PDA (doi:10.1038/nm.4314).
>> We agree that other cells in the tumor microenvironment express galectins which can affect esophageal cancer progression. Apart from the reference provided by the reviewer, there is extensive work showing expression of galectins in e.g., immune cells, fibroblasts, endothelial cells, which are all present in the tumor microenvironment. For example, we extensively published in galectins in the tumor vasculature in the past. However, we feel that an extensive analysis of this expression is beyond the scope and focus of this review, in particular because the expression of galectins in the tumor environment of esophageal cancer is not studied. Just adding evidence from other cancer types will not help the readership to understand what the role is in the context of esophageal cancer.
We already stated the following in the discussion:
"In addition, there is ample evidence that the extracellular milieu contributes to esophageal cancer progression (Palumbo et al 2020). However, whether and how this is related to galectin expression in the microenvironment of esophageal cancers requires further investigation."
For clarity, we have now added the following paragraph to make readers aware of the potential role of galectins in the tumor microenvironment (and the lack of information on this aspect in esophageal cancer:
"In addition, there is ample evidence that the extracellular milieu contributes to esophageal cancer progression (Palumbo et al 2020). Moreover, as nicely reviewed by Elola et al., galectins are known to be expressed by non-malignant cells that reside in the tumor microenvironment, including immune cells, endothelial cells, and fibroblasts (Elola et al., 2018). This expression can contribute to tumor progression by stimulating tumor cell growth, tumor angiogenesis, immunosuppression, and tumor metastasis. Thus, it can be anticipated that galectin expression and secretion in the tumor microenvironment of esophageal cancer also contributes to disease progression. However, the exact role of galectins in the microenvironment of esophageal cancers is currently unknown and requires further investigation.
(2) The author simply listed several galectins including galectin-1, galectin-3, galectin-7, galectin-9 without in-depth analysis and outlook. Specially, most studies about functions of galectins in esophageal were inconsistent and contradictory. So, will galectins be a good target for cancer therapy?
>> Indeed, based on the limited and diverse information in the literature, it is difficult to get a consistent image of the exact role of galectins in esophageal cancer. At the same time, the limited information makes it hard to predict whether or not galectins will be a good target for treatment of these patients. The main message is therefore that there are still a lot of unanswered questions that should be addressed in order to determine whether galectins are potential therapeutic targets. Given the tumor-promoting activities of galectins in other cancers and their availability on the cell surface and in the microenvironment, we do think that they might be targets for therapy. In line with comment 2 from reviewer 1, we have now more often added our personal view on such aspects in the revision.
(3) In last paragraph, authors showed “several galectin-targeting compounds are available, some of which are already evaluated in clinical trials”. But the information about clinical trials is a little, can the authors describe it in more detail as it is very important for further study of galectins.
>> In line with reviewer 2, we have expanded the paragraph on ongoing clinical trials with galectin-targeting compounds as follows:
Nowadays, several galectin-targeting compounds are available, some of which are al-ready evaluated in clinical trials in a variety of cancer types, including, colon, lung, breast, head&neck, prostate, and melanoma (Wdowiak et al., 2018; Martin-Saldaña et al., 2022). Most trials involve phase 1/2 studies to test the safety of glycan-based or small molecule inhibitors that target galectin-1 and/or galectin-3 (For a recent overview, see Martin-Saldaña et al. (Martin-Saldaña et al., 2022)). In addition, an ongoing phase1/2 trial in patients with metastasized solid tumors explores the safety, pharmacokinetics, and anti-tumor activity of a galectin-9-targeting antibody, either as monotherapy or combined with chemotherapy or anti-PD-1 treatment (NCT04666688). Thus, the upcoming years will give insight in the applicability and efficacy of these treatments. Obviously, to determine whether galectin-targeted therapies might also be successful in esophageal cancer, it is vital to gain better insights in the expression and exact role of galectins in this malignant disease. With this review, the first step in that direction was taken by highlighting the gaps in the current knowledge and identifying the outstanding challenges.
(4) There are still a few minor issues worth fixing: “in vitro and in vivo” should be italic.
>> Done.
Round 2
Reviewer 1 Report
Reviewer accept the revised version.
Reviewer 4 Report
The authors answered my questions clearly. I hope to publish their article on Cancers.